# Therapeutic Targeting of ALK in Neuroblastoma: Experience of Italian Precision Medicine in Pediatric Oncology

**DOI:** 10.3390/cancers15030560

**Published:** 2023-01-17

**Authors:** Fabio Pastorino, Mario Capasso, Chiara Brignole, Vito A. Lasorsa, Veronica Bensa, Patrizia Perri, Sueva Cantalupo, Serena Giglio, Massimo Provenzi, Marco Rabusin, Elvira Pota, Monica Cellini, Annalisa Tondo, Maria A. De Ioris, Angela R. Sementa, Alberto Garaventa, Mirco Ponzoni, Loredana Amoroso

**Affiliations:** 1UOSD Laboratory of Experimental Therapies in Oncology, IRCCS Istituto Giannina Gaslini, 16147 Genova, Italy; 2Dipartimento di Medicina Molecolare e Biotecnologie Mediche, Università degli Studi di Napoli Federico II, Via Pansini 5, 80131 Napoli, Italy; 3CEINGE Biotecnologie Avanzate, Via G. Salvatore, 486, 80145 Napoli, Italy; 4UO Pediatria-Neonatologia/Nido PO A. Ajello ASP Trapani, 91100 Trapani, Italy; 5Pediatric Oncology, Ospedale Papa Giovanni XXIII, Piazza Organizzazione Mondiale Sanità 1, 24127 Bergamo, Italy; 6Department of Pediatrics, Institute for Maternal and Child Health, IRCCS Burlo Garofolo, Via dell’Istria 65/1, 34137 Trieste, Italy; 7UOSD di Ematologia ed Oncologia Pediatrica, Università Degli Studi Della Campania “Luigi Vanvitelli,” Piazza Luigi Miraglia 2, 80138 Napoli, Italy; 8Division of Paediatric Hemato-Oncology, University Hospital Azienda Policlinico di Modena, Via del Pozzo 71, 41124 Modena, Italy; 9Department of Hematology-Oncology, Anna Meyer Children’s Hospital, VialePieraccini 24, 50139 Firenze, Italy; 10Department of Paediatric Haematology/Oncology, and Cell and Gene Therapy, Bambino Gesù Children’s Hospital, IRCCS, 00165 Rome, Italy; 11Dipartimento di Patologia, IRCCS Istituto Giannina Gaslini, 16147 Genova, Italy; 12UOC Oncologia, IRCCS Istituto Giannina Gaslini, Via Gerolamo Gaslini 5, 16147 Genova, Italy

**Keywords:** anaplastic lymphoma kinase, target therapy, tyrosine kinase inhibitor, whole-exome sequencing, neuroblastoma, precision medicine, relapsed disease, pediatric oncology

## Abstract

**Simple Summary:**

Approximately 50% of high-risk neuroblastomas (NB) relapse within two years after the end of treatment. The prognosis for relapsed or refractory patients is poor, and additional therapeutic options are needed. The identification of *ALK* somatic mutations or amplification plays an important role in the treatment of relapsed/refractory patients. The aim of our study was to evaluate the genomic status of patients with relapsed/refractory NB and to employ ALK Tyrosine Kinase Inhibitors (TKIs) in patients with targetable ALK mutations. In the era of precision medicine, ALK inhibitors may play an important role in the treatment of high-risk, ALK-mutated, NB patients.

**Abstract:**

Neuroblastoma (NB) is the most common extracranial solid tumor in childhood. Patients with relapsed/refractory disease have a poor prognosis, and additional therapeutic options are needed. Mutations and amplifications in the *ALK* (Anaplastic Lymphoma Kinase) gene constitute a key target for treatment. Our goal, within the Italian project of PeRsonalizEdMEdicine (PREME), was to evaluate the genomic status of patients with relapsed/refractory NB and to implement targeted therapies in those with targetable mutations. From November 2018 to November 2021, we performed Whole Exome Sequencing or Targeted Gene Panel Sequencing in relapsed/refractory NB patients in order to identify druggable variants. Activating mutations of *ALK* were identified in 8(28.57%) of 28 relapsed/refractory NB patients. The mutation p.F1174L was found in six patients, whereas p.R1275Q was found in one and the unknown mutation p.S104R in another. Three patients died before treatment could be started, while five patients received crizotinib: two in monotherapy (one with p.F1174L and the other with p.S104R) and three (with p.F1174L variant) in combination with chemotherapy. All treated patients showed a clinical improvement, and one had complete remission after two cycles of combined treatment. The most common treatment-related toxicities were hematological. ALK inhibitors may play an important role in the treatment of ALK-mutated NB patients.

## 1. Introduction

Neuroblastoma (NB), a childhood cancer of the developing sympathetic nervous system, is the most common extracranial pediatric solid tumor, accounting for 8–10% of childhood malignancies and 15% of pediatric oncology deaths [1]. Heterogeneity is a clinical hallmark of NB, which displays a wide range of clinical behaviors and diverse responses to treatments [2]. While a subgroup of tumors may present spontaneous regression and a good prognosis, patients with metastatic disease, which is classified as a high-risk (HR) NB, have a poor prognosis despite the use of intensive multimodal approaches [2,3]. Newly diagnosed HR-NB patients have an overall survival (OS) rate of less than 50% [3,4,5,6,7]. Furthermore, HR-NB patients are characterized by high rates of relapse after first-line treatment (approximately 50%); this constitutes a major clinical problem owing to the difficulty of treating relapsing disease [8,9]. Precision medicine is an expanding field and the molecular characterization of cancer can provide prognostic information and identify targets for molecular therapy [10,11,12].

Genomic sequencing studies have shown that NB, like other pediatric cancers, presents fewer recurrent somatic mutations, enriched at relapse [13,14]. Full characterization of genetic alterations is crucial to improving treatment and outcome. In NB, mutations and amplifications in the *ALK* (Anaplastic Lymphoma Kinase) gene correlate with an adverse prognosis, even in patients classified as being at low or intermediate risk, owing to increased *ALK* mRNA and protein expression [15,16,17,18,19].

The tyrosine kinase ALK, a member of the insulin receptor tyrosine kinase family, was first discovered more than 17 years ago in fusion with nucleophosmin (NPM) in a subset of anaplastic large-cell lymphomas (ALCL) [20]. The *ALK* oncogene is aberrantly expressed in many adult cancers, mainly in non-small-cell lung cancer (NSCLC) [21] and anaplastic large-cell lymphoma (ALCL) [22]. It is also common in pediatric cancers such as ALCL, inflammatory myofibroblastic tumors (IMT) and NB [23].

ALK receptor tyrosine kinase can be constitutively activated either by chromosomal translocations, leading to ALK-fusion proteins, or by point mutations. Somatic activating mutations have been reported in 10–12% of sporadic NB cases, while germline *ALK* mutations have been described in 1–2% of familial NB cases [24]. On relapse, the incidence of *ALK* point mutations increases to about 25% [13,25]. This dynamic event highlights the importance of tumor re-biopsy and repeated *ALK* state evaluation during disease progression. The most frequent mutations observed are *ALK*-R1275 (43%), *ALK*-F1174 (30%) and *ALK*-F1245 (12%), while *ALK*-I1171N and *ALK*-Y1278S [15,18] are less frequent. *ALK* gene amplification is also found in a small percentage (about 5%) of primary tumors [26]. Biochemical and computational studies have distinguished oncogenic mutations (constitutively activating) from non-oncogenic mutations and have highlighted the sensitivity of mutated variants to targeted therapies [18]. ALK-mutated NB may benefit from tumor-targeted therapies with ALK tyrosine kinase inhibitors (TKI). Indeed, ALK TKI have been employed in preclinical and clinical studies to assess the efficacy and safety of ALK inhibition in cancer cells harboring *ALK* point mutations [18,27,28]. In 2011, the ATP-competitive small molecule crizotinib was approved as an ALK inhibitor for the treatment of ALK-positive NSCLC [29].

Through the implementation of precision oncology and the use of high-throughput sequencing within the Italian protocol of personalized medicine (PREME) for NB [12], we explored the possibility of applying precision medicine to the management of refractory/relapsed NB patients by discovering actionable *ALK* alterations.

## 2. Materials and Methods

Patients with refractory/relapsed neuroblastoma (NB) for whom there was no known curative treatment were eligible in the PREME. Participants provided written consent to the clinical protocol after the risks and benefits had been explained to them and/or their caregivers. The protocol envisioned the potential disclosure of medically actionable secondary findings, defined as germline and somatic mutations unrelated to the condition for which sequencing was being performed. Specifically, patients could choose whether or not to consent to following: disclosure of the primary and/or secondary results of the study; having their samples and/or data stored for future research. All procedures were performed in accordance with relevant guidelines and regulations and approved by the CER (Regional Committee), first under the protocol ANTECER_Neuroblastoma: 15/12/2016, amendment 065_16/09/2019, and subsequently under the protocol EudraCT: 2022-000558-27.

Patients had to have confirmed *ALK* translocations, activating mutations, or amplification to receive crizotinib treatment. Patients received crizotinib at the dose of 280 mg/m^2^/dose given orally, twice daily, on a continuous schedule in cycles of 28 days duration, according to Children’s Oncology Group (COG) study ADVL0912 [27].When combined with chemotherapy, crizotinib was administered orally, twice daily, at the dose of 215 mg/m^2^/dose, on days 1–21 of each 21-day cycle associated with cyclophosphamide (at the dose of 250 mg/m^2^/dose)and topotecan (0.75 mg/m^2^/dose) on days 1–5 of each cycle [30].

### 2.1. Sample Preparation and Characterization

Biological samples (NB tumor tissues, bone marrow (BM)-infiltrating NB cells and peripheral blood) were centralized at the Laboratory of Experimental Therapies in Oncology, IRCCS Istituto Giannina Gaslini. BM-infiltrating NB cells and peripheral blood were collected in EDTA and freshly used for flow cytometry immunophenotyping and DNA extraction.

### 2.2. NB Tumor Characterization

Samples taken from 17 NB tumors on relapse were kept in culture medium (RPMI-1640) for 24 h or stored in a freezing solution containing 90% of serum and 10% DMSO until use (histological immunophenotyping and DNA extraction). After paraffin embedding, immunohistochemistry phenotyping was carried out. Tumor tissue sections (3 µm) were de-paraffinized and incubated with the immune-cell marker CD45 (LCA) (Mouse Monoclonal Antibody (mAb)—Cell Marque/Roche), and with the NB markers CD56 (MRQ-42) (Rabbit mAb—Cell Marque/Roche), TH (F-11) (Mouse mAb—Santa Cruz Biotechnology), PHOX-2B (EPR14423) (rabbit mAb—Abcam) and S100 (4C4.9) (Mouse Monoclonal Antibody—Ventana/Roche). In some cases, primary Abs were diluted with Ventana/Roche Ab diluent (for TH mAb) or with BOND Primary Antibody Diluent—Leica (for PHOX-2B mAb). The ultraView Universal DAB detection kit from Ventana and the BOND Polymer Refine Detection kit from Leica (for PHOX-2B mAb) were used to detect the binding of primary antibodies. Sections were counterstained with HEMATOXYLIN—Ventana/Roche.

### 2.3. Characterization and Enrichment of BM-Infiltrating NB Cells

NB-infiltrated BM samples taken from 11 NB patients on relapse underwent immunophenotyping. Briefly, 60 µL of whole BM blood was dispensed into each tube and stained with the following monoclonal antibodies (mAbs): a-CD45 PE-Cy7 (mouse IgG1, clone 2D1; Invitrogen), a-CD56 APC (mouse IgG1, clone CMSSB; Invitrogen), a-GD2 PE (mouse IgG2a, clone 14G2a; Biolegend), a-B7-H3 PE (mouse IgG1, clone DCN.70; Biolegend), a-NCL AlexaFluor 488 (mouse IgG1, clone 364-5; Abcam). Samples were incubated for 25 min, at 4 °C. Red cells were then lysed by means of 1× PharmLyse, according to the manufacturer’s instructions. After a final washing step with PBS (1% FBS, 2 mM EDTA), samples were analyzed by means of flow cytometry (FCM). The percentage of NB cells infiltrating the BM was defined as CD45neg/CD56pos, and either GD2pos or/and B7-H3pos.

BM samples displaying 1–50% of NB cell infiltration underwent subsequent NB cell enrichment. The percentage of NB cells infiltrating the BM is directly calculated by the Kaluza software (Beckman Coulter Life Sciences), which is used to analyze the stained BM samples. Specifically, NB cells are defined as CD56/GD2/B7-H3 positive cells within the CD45 negative population. Based on this analysis, the software determines the percentage of NB cells on the total number of cells.

BM samples underwent immune-magnetic manipulation in order to positively separate labeled NB cells through the use of magnetic microbeads. Specifically, NB cells infiltrating the BM samples were incubated with human FcR blocking reagent (MiltenyiBiotec) for 10 min at 4 °C, to increase the specificity of immunofluorescent staining, and then labeled with either anti-GD2 PE or anti-B7-H3 PE mAbs. The decision of whether to use anti-GD2 or anti-B7-H3 mAbs was based on the best expression of the respective antigens, as revealed by immunophenotyping. After being stained, cells were incubated with anti-mouse IgG microbeads, according to the manufacturer’s instructions (Miltenyi Biotec). MS or LS MACS separation columns (Miltenyi Biotec) were used to collect the fraction of labeled NB cells, as previously described [31,32]. At the end of this procedure, the positively separated fraction of NB cells (GD2pos or B7-H3pos) was further analyzed in order to determine the percentage of CD45pos cells contaminating the samples. After NB cell enrichment, at least 8 × 10^5^ NB cells were used for DNA extraction.

### 2.4. DNA Extraction and Sequencing

DNA was extracted from tumor tissues, BM-infiltrating NB cells, and also from peripheral blood of the corresponding patients to obtain constitutional DNA. The QIAamp DNA mini kit was used according to the manufacturer’s instructions (QIAgen). The kit is designed to purify total DNA by means of columns containing a silica-based membrane. A RNase A treatment was included during DNA extraction to avoid contamination of RNA in the NGS downstream applications.

DNA samples were stored at −20 °C until use. Genomic sequencing analysis was carried out by means of whole-exome sequencing (WES, 100× coverage) for samples with tumor cellularity ≥ 70%, or deep targeted gene panel sequencing (TGPS, 1000× coverage) for samples with tumor cellularity ranging between 20% and 70%. Libraries were generated by means of Agilent Sure Select Human Whole Exon Kit v.6. Paired-end sequencing (2 × 150 bp) was performed by means of the IlluminaHiSeq 1500 platform. Sequencing data (tumor tissue and control data (reads)) were aligned against the human reference genome (GRCh37/hg19) by means of the Burrows–Wheeler Aligner (BWA) program. The alignment files were sorted by genomic coordinates by means of the Sam tools program; the same program was used to remove duplicate reads (aligned with identical genomic coordinates and resulting from DNA PCR amplification steps). Tumor tissue somatic variants (single nucleotide variants, SNVs, and small (up to 50 bp) insertions/deletions, INDELs) were identified by means of the GATK Mutect2 program. Finally, the lists of identified variants were annotated by means of ANNOVAR.

### 2.5. Data Interpretation and Reporting

Interpretation of clinical WES and TGPS was conducted via the Molecular Tumor Board (MTB), and approximately 30 days after the request for testing, a report was generated. For the purpose of clinical discussion, the report included variants with a pathogenicity score of at least 20 (calculated by CADD) and a CancerVar score greater than 0.80 in genes that potentially interact with anti-neoplastic drugs according to the DGIdb database (https://www.dgidb.org, 10 January 2023). The Therapeutic Target databases (dB) were also used, in order to highlight “potential target genes”.

## 3. Results

From November 2018 to November 2021, 29 samples from 28 patients with relapsed/refractory NB were collected. To examine the spectrum of *ALK* mutations in our cohort, we analyzed germline and somatic *ALK* point mutations in blood, BM-infiltrating NB cells and NB tumor tissue samples.

Activating mutations of *ALK* were identified in eight patients (28.57%): four relapsed and four refractory patients with a median age of 6 years (range:4–22 years) on inclusion.

The characteristics of the eight patients with *ALK* aberrations are summarized in Table 1.

Molecular profiling was performed by using tumor tissue in three cases and BM-infiltrating NB cells in five cases. The median time from tumor tissue collection to MTB recommendations was 43 days (range: 31 to 96 days), including the time for tumor characterization, DNA preparation, sequencing and bioinformatic analyses.

Only four of the eight patients with *ALK* mutations had *MYCN* amplification. The same *ALK* point mutation (F1174L) was detected in six patients; a rare mutation (S104R) and the more common mutation (R1275Q) were found in the other two patients (Table 2).

All cases, except for that of one patient who died during the sequencing analysis, were discussed in the MTB by experts in NB, experts in new drugs, a geneticist, biologists and the treating physician; during this discussion, the patient’s history and other treatment options were considered.

Combination therapy with TKI and backbone chemotherapy was administered in three patients.

Patient L, who was treated with the combination of crizotinib–topotecan–cyclophosphamide, showed a CR after the first 2 cycles, remaining alive and in complete remission for 8 months of treatment; however, therapy was discontinued owing to transplant-related mortality (TRM) after haploidentical hematopoietic stem cell transplantation to consolidate the response obtained. The most frequent and serious adverse event was hematological toxicity. Grade 3 neutropenia and thrombocytopenia occurred after the first and second cycles. Crizotinib was administered at the same dose as in the first two cycles, while the doses of both drugs (Topo-Cyclo) were reduced by 25% in subsequent cycles, during which hematologic tolerance was good.

Patient 21 also showed a clinical response to the combination of crizotinib–topotecan–cyclophosphamide. However, grade 2–3 hematological and gastrointestinal toxicity prompted periodic discontinuation of treatment. Cycle 1 of crizotinib–Topo–Cyclo was complicated by Clostridium enteritis and fever. After radiotherapy at the site of relapse (30 Gy), crizotinib was resumed in combination with oral cyclophosphamide and etoposide; the patient’s clinical condition remained stable for 7 months.

Patient 20, who was treated with TEMIRI (Temozolamide 100 mg/m^2^/dose and Irinotecan 50 mg/m^2^/dose on days 1–5, respectively) and Crizotinib, showed a mixed response. After the first cycle, CR of her liver metastasis was observed; however, the rapid progression of bone disease led to death during the second cycle.

The MTB initially deemed patient N (Pt N) ineligible for targeted treatment with ALK inhibitor, owing to the rare mutation found (S104R), the pathological significance of which was uncertain. The patient was therefore treated with 4 cycles of second-line chemotherapy (Temozolamide-Irinotecan), which achieved a PR, and then enrolled in a phase II immunotherapy trial. However, the disease progressed. In view of the lack of alternative treatment, we decided to treat the patient with crizotinib. During the first 21 days of crizotinib administration, her clinical status improved, and morphine use and supportive care were reduced. However, the patient died of PD.

Patient M received two cycles of crizotinib, which elicited a radiometabolic PD. For this reason, he switched to Lorlatinib for four cycles, reaching a state of stable disease. To achieve a greater radiological response, Lorlatinib and oral cyclophosphamide were combined. The patient received three cycles of this combination without toxicity. At the last assessment, the disease was seen to have remained stable. Treatment is still ongoing.

## 4. Discussion

Children and adolescents with relapsed or refractory neuroblastoma (NB) have a particularly poor prognosis. Survival rates of less than 25% following recurrence suggest an urgent need for innovative treatment strategies, such as precision medicine or personalized treatments.

Advances in genomic technology and targeted therapeutics are increasingly contributing to the development of precision medicine. The goal of precision medicine in pediatric oncology is to develop more effective and less toxic therapies. Several recent studies on precision medicine in pediatric oncology have demonstrated the potential of targeted therapy [12,33,34,35], in that the majority of patients involved have at least one actionable genetic alteration. In many pediatric cancers, a clearly defined genomic target is lacking. The enrichment of targetable mutations in the relapsed NB genome confirms the importance of collecting and analyzing relapsed/refractory tissue.

So far, however, the clinical integration of genome sequencing into standard clinical practice has been limited, and many obstacles remain. As resistance to single targeted therapies constitutes a serious limit, the power of the synergic effect of standard chemotherapy offers a precious opportunity. The PREME protocol applies molecular profiling in order to provide information on actionable gene variants, which may be used for clinical trial enrollment or experimental approaches (compassionate or off-label use).

Mossè et al. found that the presence of *ALK* aberration correlated with lower survival in patients with HR-NB [27]. In this regard, ALK inhibitors, such as crizotinib, have provided novel treatment opportunities [27,36].

The safety and efficacy of crizotinib was tested in a phase I/II study involving pediatric patients with recurrent and refractory solid tumors (ADVL0912) [27]. The drug was well tolerated as a single agent, eliciting an objective tumor response in 3 out of 11 NB ALK-mutated patients (2 stable disease (SD) and 1 complete response (CR)). In NB patients with unknown *ALK* status, an objective response was documented in 6/23 (1 CR and 5 SD). The recommended phase 2 dose of crizotinib as single agent was 280 mg/m^2^/dose [27]. However, long-term treatment with crizotinib and other TKIs is limited by the development of drug resistance [27,37]. Some studies have reported that the point mutations F1174L and F1245C are the most frequent in crizotinib-resistant NB [38]. Single-target agents are rarely sufficient to obtain remission or a durable response in NB patients, owing to the acquisition of secondary mutations, which lowers the binding affinity of the inhibitors, and pathway evasion strategies of the cancer cells. In preclinical studies, crizotinib in combination with backbone chemotherapy has displayed synergistic activity in NB cell lines with the most common *ALK* mutations. Indeed, in vitro and in vivo studies have shown that the combination of crizotinib and chemotherapy can overcome crizotinib resistance [39,40]. Clinical studies have also tested crizotinib and other TKIs in association with standard chemotherapy [30,40].

However, *ALK*-F1174L mutations result in de novo crizotinib resistance, while *ALK*-R1275Q is the point mutation with the highest sensitivity [39].

Bresler et al. [18] found that *MYCN* amplification occurred more frequently (39%) than the expected overall frequency (21%) in *ALK*-F1174 mutated patients. Patients with both amplified *MYCN* and *ALK*-F1174 mutation had significantly worse event-free survival.

One option that has been proposed is to combine crizotinib with chemotherapeutic agents commonly used for relapsed NB [30,40,41]. Preliminary data suggest that this combination therapy can overcome crizotinib resistance.

From a mechanistic and functional point of view, the worsening outcome of NB patients with tumors carrying both *ALK* mutations and *MYCN* amplification can be explained by promoter analysis revealing *ALK* as a transcriptional target of *MYCN* [17] Therefore, amplified/overexpressed *MYCN* leads to increased levels of ALK in whichever form, either wild-type or mutated. In turn, deregulated ALK generates a positive feedback loop by inducing an ERK5-mediated *MYCN* transcription through the PI3K, AKT, MEK3 and MEK5 signaling pathways, thus potentiating the oncogenic activity of MYCN [42].

In a phase I consortium study (ADVL1212) involving a pediatric population with recurrence/refractory solid tumors, Greengard et al. analyzed the safety and efficacy of crizotinib in combination with chemotherapy (topotecan/cyclophosphamide (Topo-Cyclo) or Vincristin/Doxorubicine (VCR/DOXO)); an objective response was documented in seven patients (15.9%): two CR and five PR [30]. The efficacy of crizotinib combined with standard chemotherapy for *ALK*-mutant HR-NB is currently being evaluated in a COG Phase III trial (NCT03126916).Owing to the limitations of the first-generation ALK inhibitors, such as crizotinib, novel drugs, such as Ceritinib, Entrectinib and Lorlatinib, have been designed [43,44,45,46].

In our study, activating mutations of *ALK* were identified in 8 of 28 relapsed/refractory NB patients (28.57%); 5 of these 8 patients were treated with crizotinib, alone or in association with chemotherapy. Indeed, ALK inhibitors, in combination with standard chemotherapy, might have the potential to induce durable remission or disease stability in patients harboring *ALK* mutations. The baseline condition of patients with a very high priority target, such as *ALK* mutations, who receive matching targeted treatment is reported to be slightly better than that of all other patients. In our cohort, all patients received second-line chemotherapy before target therapy. In NB patients, molecular diagnostics should be implemented at an earlier stage during their disease course and not only at a stage when precision medicine is “the last hope”.

One of the main limitations of precision medicine is that even though actionable alterations have been found, drug availability is limited. Clinicians are therefore obliged to request drugs for compassionate or off-label treatment. Another important limitation is that the use of compassionate or off-label treatment prevents the drafting of a real protocol, responding to the need for personalized therapy. In conclusion, efforts are ongoing to increase the clinical trials available for children with relapsed or refractory disease. However, in NB patients with targetable mutations, consistent protocol should be used to parse out the effects of the TKI used.

## 5. Conclusions

Therapeutic targeting of the ALK kinase is a promising strategy for the approximately 20% of NB patients. Targeted therapies have made great strides in the treatment of aggressive malignancies such as NB. However, tumor cells develop mechanisms of resistance to even the most specific agents. More research is needed to identify the most effective ALK inhibitors in combination with standard chemotherapy. 

ALK inhibitors combined with chemotherapy could have the potential to induce a major response or durable remission for NB patients harboring *ALK mutations*. 

## Figures and Tables

**Table 1 cancers-15-00560-t001:** Characteristics of patients and targeted treatment recommendations for *ALK* alterations.

Pt Code PREME	Sex	Age on Diagnosis (Years)	Stage/ *MYCN*	Relapse/ Refractory	Age on Relapse or Progression (Years)	Therapy Recommendation	Follow-Up
Pt#3	M	4	M/ ampl	Relapse	5	Ceritinib	Died before starting treatment
Pt#5	M	4	M/ non-ampl	Relapse	6	Crizotinib, ceritinib, alectinib	Died before starting treatment
Pt#19	F	3	M/ ampl	Refractory	4	/	Died before WES report
Pt#20	F	2	M/ ampl	Refractory	5	Crizotinib–TEMIRI	After 1 cycle, CR liver metastasis but bone PD Died after 2 cycles
Pt#21	F	17	M/ non-amp	Refractory	22	Crizotinib– Topo–Cyclo	Alive with SD, on treatment at 7 months
Pt#L	M	3	M/ ampl	Relapse	7	Crizotinib– Topo–Cyclo	CR after 2 cycles, CR persisted for 8 months of treatment. Died of TRM
Pt#M	M	10	M/ non-ampl	Refractory	15	Crizotinib	Alive with SD, after 2 cycles of crizotinib and 7 cycles of lorlatinib
Pt#N	F	5	M/ non-ampl	Relapse	8	Crizotinib	Died after 1 cycle

Abbreviations: Pt: patient; PREME: PeRsonalizEdMEdicine; amp: amplified; WES: whole-exome sequencing; TEMIRI: temozolomide + irinotecan; Topo: topotecan; Cyclo: cyclophosphamide; SD: stable disease; CR: complete response; PD: progressive disease; Stage M: distant metastatic disease; TRM: transplant-related mortality.

**Table 2 cancers-15-00560-t002:** Characteristics of patients: molecular profiling results.

Pt Code PREME	Variant	Total Depth	Fraction (%)	SNV ID	COSMIC	CADDv16 Score	CancerVar
Pt#3	c. C3522A p. F1174L	137x	34	rs863225281	COSM28055	29	0.96
Pt#5	c. G3824A p. R1275Q	310x	6	rs113994087	COSM28056	35	0.96
Pt#19	c.C318A p. F1174L	201x	27	rs863225281	COSM28055	29	0.96
Pt#20	c.C318A p.F1174L	160x	51	rs863225281	COSM28055	29	0.96
Pt#21	c.C318A p. F1174L	151x	26	rs863225281	COSM28055	29	0.96
Pt#L	missense P. F1174L	3364	39	rs863225281	COSM28055	29	0.96
Pt#M	c.C318A p. F1174L	198x	19	rs863225281	COSM28055	29	0.96
Pt#N	^ c.C312A p.S104R	149x	25	-	-	26.7	0.45

Pt: patient; PREME: PeRsonalizEdMEdicine; SNV: Single Nucleotide Variant; ^ hg19_ Chr:2 Pos:29443701 G>T.

## Data Availability

No new data were created.

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
