# Peer review of "Therapeutic Targeting of ALK in Neuroblastoma: Experience of Italian Precision Medicine in Pediatric Oncology"

_cancers, 2023, doi:10.3390/cancers15030560_

Round 1
Reviewer 1 Report
Authors explained the Therapeutic targeting of ALK in NB, especially in refractory/relapsed NB. Although ALK has been reported in multiple cancers, including NB previously.
Minor Comments:
1. Please correct or rephrase lines 69-72
2. Extensive English editing is required as some of the sentences are confusing.
Author Response
We thank the Reviewer for the kind comments.
Accordingly to the Reviewer, we rephrased lines 69-72 and we performed an extensive English editing.
Reviewer 2 Report
Pastorino F., Capasso M. et al. describe in the manuscript “Therapeutic targeting of ALK in neuroblastoma: experience of the Italian Precision Medicine in Pediatric Oncology” molecular genomic profiling and ALK inhibitor therapy in a cohort of 28 Italian patients with refractory and relapsed neuroblastoma. The authors demonstrate that 8 patients were identified, 7 with known activating mutations in the ALK tyrosin kinase domain and 1 patient with a so far undescribed ALK mutation. Five patients received ALK inhibitor treatment with Crizotinib alone or in combination with chemotherapy. All patients showed clinical improvement under therapy.
Major:
1. While the paper is well written and an interesting report of the Italian experience with precision oncology diagnostics and ALK inhibitor therapy, there are no significant new aspects, since seminal papers on ALK alterations in large cohorts and ALKi treatment (Crizotinib, Ceritinib, Lorlatinib) in the refractory/relapse setting have been published in recent years, e.g. PMID: 34780709, PMID: 34115544, PMID: 33568345, PMID: 23598171.
2. The methods section lacks details, for example it is not clear how neuroblastoma cells were isolated from infiltrated bone marrow. It is recommended that the authors either refer to a published protocol or describe the procedure (including the antibody used etc). Furthermore, it is not specified whether blood was used was a source for genomic DNA or cell-free tumor DNA was isolated from peripheral blood plasma.
Minor:
1. English proof-reading is required. There are some text formatting problems as well.
2. Citations are missing at times (e.g. lines 226 – 228)
Author Response
We agree with the Reviewer that there are no significant new scientific aspects, indeed our study wants to represent the Italian experience of precision oncology by using of ALK inhibitor therapy, as an example of “real-life” application of personalized medicine.
As requested by the Reviewer, we implemented methods, describing isolation of tumor cells from infiltrated BM, and we provided a more complete overview on all papers focusing on ALK mutations in NB.
We performed an extensive English editing and we updated the citations
Reviewer 3 Report
Interesting concept for a possible treatment for individuals with neuroblastoma refractory to treatment. However, there are a number of points the authors need to address in order to .
1. The actual protocol followed for treating patients with crizotinib should be clearly described in the methods section (eg dose, time interval between 1st and 2nd injection, chemotherapeutic agent(s) given (dose and time intervals). This information is critical as it appears that not all patients were treated using the same protocol and since the number of patients studied was small, it is difficult to know what was actually influencing a response.
2. The time span over which the responsive patients were followed may not have been sufficient to ascertain whether they were completely free of neuroblastoma. This is especially true for those with stable disease.
3. The limited number of patients makes any conclusions difficult. Need to study more patients that meet the criteria used, treat each with the same protocol, and if a patient appears NB free, follow them over at least a two year period. Those with stable disease also need to be followed to ascertain how long and whether treatment needs to be ongoing.
4. Did all of the patients studied present with NB in the same primary site?
5. What causes resistance to the ALK-inhibitor crizotinib?
6. Line 67 states: Genetic sequencing studies have shown that NB, as well as other pediatric cancers present fewer somatic mutations, enriched at relapse. Should it say that ....present fewer somatic mutations than those at relapse?
7. At the end of the discussion the authors indicate that more patients need to be studied. Included in that statement should be that a consistent protocol should be used is they want to parse out the effects of the TKI used.
Author Response
- As requested by the Reviewer, treatment indications have been added in the methods section.
2 and 3. As requested by the Reviewer, two paragraphs have been added in the Discussion section.
Another important limitations is that the use of compassionate or off-label treatment prevents the drafting of a real protocol, responding to the need for personalized therapy. Furthermore, these patients have a short life expectancy, the follow-up is sometimes too short to evaluate side effects and long-term efficacy.
In conclusion, efforts are ongoing to increase the clinical trials available for children with relapsed or refractory disease. However, in NB patients with targetable mutations, consistent protocol should be used to parse out the effects of the TKI used.
- Did all of the patients studied present with NB in the same primary site?
AAs: The most frequent site of relapse is bone and bone marrow. In our cohort of 28 patients, 60% presented a metastatic relapse, 30% a refractory disease and only 10% a local relapse.
- What causes resistance to the ALK-inhibitor crizotinib?
AAs: The causes can be various: specific ALK mutations, overexpression of IL10RA in the IL-10 signaling pathway, increased STAT3 activity (see: Camidge DR. Next-generation ALK inhibitors: is the median the message? Lancet Respir Med (2020) 8(1):5–7. doi: 10.1016/S2213-2600(19)30362-5. Hu G, Feldman AL. Drivers of crizotinib resistance in ALK+ ALCL. Blood (2020) 136(14):1573–5. doi: 10.3390%2Fcancers13236003).
- Line 67 states: Genetic sequencing studies have shown that NB, as well as other pediatric cancers, present fewer somatic mutations, enriched at relapse.
AAs: We thank the Reviewer and the phrase has been changed accordingly.
“At relapse, pediatric cancers present more somatic mutations than at first diagnosis.”
- At the end of the discussion the authors indicate that more patients need to be studied. Included in that statement should be that a consistent protocol should be used is they want to parse out the effects of the TKI used.
AAs: Again, we thank the reviewer for the correct comment and the conclusion of the discussion section has been changed accordingly.
Reviewer 4 Report
High-risk neuroblastoma is a major problem. In this article authors discussed there experience in treating high-risk neuroblastoma patients with ALK mutations and recommend performing sequencing during diagnostic stage to identify therapeutic targets. Authors also pointed that chromosomal alterations or mutations enrich at recurrence, therefore recommend sequencing at multiple intervals to be part of standard treatment regimen. This article is well written. Articles like this will help Neuroblastoma community in developing novel therapies. Appreciate the authors for their contribution. Article has space missing between words in most areas. I recommend revising manuscript throughly for formatting errors.
Author Response
We thank the Reviewer for the kind comments.
Round 2
Reviewer 3 Report
Authors have responded to essentially all of this reviewer's previous comments. The one exception is that while they indicate that a change was made, it was not incorporated in the revised manuscript: See line 72: Genomic sequencing studies have shown that NB, like other pediatric cancers, presents fewer recurrent somatic mutations, enriched at relapse should read towards the end: ....somatic mutations when first diagnosed than at relapse.
Points for clarification
Paragraph starting at line 358: when discussing MYCN and ALK, a sentence or two discussing the metabolic relationship between them would be helpful.
About line 395: If the drug is working, the patient's life expectancy should be lengthened to the extent that side effects could be evaluated. If the life expectancy is so short that the effects cannot be measured, is the drug really effective?
Line 392: limitations should be singular, not plural.
Line 128, what does mq mean when giving the dose as 280 mg/mq? In subsequent statements in the same paragraph dose is given as mg/m2.
Lines 172-174 when discussing percentage of NB cells infiltrating the BM: it is not readily apparent on how the percentage was calculated. Basically you need state that you took the number of NB cells divided it by the total number of all cells and multiplied by 100.
Line 191 when discussing the number of NB cells used for extraction of DNA, the 5 in 8 X 105 should be an exponent.
Author Response
We thanks the Reviewer for the accurate revision of our manuscript giving us the opportunity to further improve the quality of the paper.
Points for clarification:
Paragraph starting at line 358: when discussing MYCN and ALK, a sentence or two discussing the metabolic relationship between them would be helpful.
We agree with the reviewer and we the following sentences (with the related references) have been added in the Discussion section:
From a mechanistic and functional point of view, the worsening outcome of NB patients with tumors carrying both ALK mutations and MYCN amplification can be explained by promoter analysis revealing ALK as a transcriptional target of MYCN [Hasan, M. K., A. Nafady, A. Takatori, S. Kishida, M. Ohira, Y. Suenaga, S. Hossain, J. Akter, A. Ogura, Y. Nakamura, et al. "Alk is a mycn target gene and regulates cell migration and invasion in neuroblastoma." Sci Rep 3 (2013): 3450. 10.1038/srep03450. https://www.ncbi.nlm.nih.gov/pubmed/24356251]. Therefore, amplified/overexpressed MYCN leads to increased levels of ALK in whichever form, either wild-type or mutated. In turn, deregulated ALK generates a positive feedback loop by inducing an ERK5-mediated MYCN transcription through the PI3K, AKT, MEK3 and MEK5 signaling pathways, thus potentiating the oncogenic activity of MYCN [Perri P, Ponzoni M, Corrias MV, Ceccherini I, Candiani S, Bachetti T. "A Focus on Regulatory Networks Linking MicroRNAs, Transcription Factors and Target Genes in Neuroblastoma." Cancers (Basel) (2021) 13(21):5528. doi: 10.3390/cancers13215528].
About line 395: If the drug is working, the patient's life expectancy should be lengthened to the extent that side effects could be evaluated. If the life expectancy is so short that the effects cannot be measured, is the drug really effective?
We agree and the not clear sentence at pag.10 of the Discussion section has been deleted.
Line 392: limitations should be singular, not plural.
Thanks, it was corrected
Line 128, what does mq mean when giving the dose as 280 mg/mq? In subsequent statements in the same paragraph dose is given as mg/m2.
Thanks, mg/mq it was corrected in mg/m2
Lines 172-174 when discussing percentage of NB cells infiltrating the BM: it is not readily apparent on how the percentage was calculated. Basically you need state that you took the number of NB cells divided it by the total number of all cells and multiplied by 100.
Thanks, we clarified as we calculated the percentage of infiltrating NB cells by adding the following sentence in the M&M pag. 4:
The percentage of NB cells infiltrating the BM is directly calculated by the Kaluza software (Beckman Coulter Life Sciences) which is used to analyse the BM stained samples. Specifically, NB cells are defined as CD56/GD2/B7-H3 positive cells within the CD45 negative population. Based on this analysis, the software determines the percentage of NB cells on the total number of cells.
Line 191 when discussing the number of NB cells used for extraction of DNA, the 5 in 8 X 105 should be an exponent.
Thanks, it was corrected
Grammar and spelling errors have been corrected throughout the paper.